# Smoking Affects the Post-Stroke Inflammatory Response of Lipid Mediators in a Gender-Related Manner

**DOI:** 10.3390/biomedicines11010092

**Published:** 2022-12-29

**Authors:** Arleta Drozd, Dariusz Kotlęga, Krzysztof Dmytrów, Małgorzata Szczuko

**Affiliations:** 1Pomeranian Medical University in Szczecin, 70-204 Szczecin, Poland; 2Department of Pharmacology and Toxicology, University of Zielona Góra, 65-417 Zielona Góra, Poland; 3Institute of Economics and Finance, University of Szczecin, 70-453 Szczecin, Poland

**Keywords:** ischemic stroke, inflammation, eicosanoids, smoking, gender

## Abstract

The main goal of our study was to determine the effect of cigarette smoking on selected derivatives of arachidonic acid, linoleic acid, DHA, and EPA, which may be markers of post-stroke inflammation. The eicosanoid profile was compared in both smoking and non-smoking patients, without division and with division into gender. In the group of non-smokers, we observed higher levels of the linolenic acid derivative (LA) 9S HODE (*p* ≤ 0.05) than in smokers. However, after dividing the results by sex, it turned out that the level of this derivative was higher in non-smoking women compared to smoking women (*p* ≤ 0.01) and did not differentiate the group of men. Similarly, the level of the arachidonic acid metabolite LTX A4 (*p* ≤ 0.05) differed only in the group of women. In this group, we also observed a decreased level of 15S HETE in smoking women, but it was statistically insignificant (*p* ≤ 0.08). On the other hand, the level of this derivative was statistically significantly higher in the group of non-smoking women compared to male non-smokers. The group of men was differentiated by two compounds: TXB2 and NPD1. Male smokers had an almost two-fold elevation of TXB2 (*p* ≤ 0.01) compared with non-smokers, and in this group, we also observed an increased level of NPD1 compared with male non-smokers. On the other hand, when comparing female non-smokers and male non-smokers, in addition to the difference in 15S HETE levels, we also observed elevated levels of TXB2 in the group of non-smokers. We also analyzed a number of statistically significant correlations between the analyzed groups. Generally, men and women smokers showed a much smaller amount of statistically significant correlations than non-smokers. We believe that this is related to the varying degrees of inflammation associated with acute ischemic stroke and post-stroke response. On the one hand, tobacco smoke inhibits the activity of enzymes responsible for the conversion of fatty acids, but on the other hand, it can cause the failure of the inflammatory system, which is also the body’s defense mechanism. Smoking cigarettes is a factor that increases oxidative stress even before the occurrence of a stroke incident, and at the same time accelerates it and inhibits post-stroke repair mechanisms. This study highlights the effect of smoking on inflammation in both genders mediated by lipid mediators, which makes smoking cessation undeniable.

## 1. Introduction

Ischemic stroke is a significant public health problem worldwide. It is the third most common cause of death and one of the leading causes of permanent disability. With the aging of the Western population and the increase in risk factors, the risk of stroke is also increasing [1]. The most common type of ischemic stroke is the atherothrombotic type, with an incidence ranging from 70% to 80% of cases [2,3].

The etiopathogenesis of stroke is complex. Epidemiological studies show differences based on gender, age, and ethnicity. On the other hand, cigarette smoking is considered to be the main modifiable risk factor for stroke, as well as for other cardiovascular diseases (CVDs). Active smokers have a two to three times higher relative risk of stroke than non-smokers [4]. The results of meta-analyses show a link between the number of cigarettes smoked and an increased risk of stroke, especially ischemic stroke. An increase in the number of cigarettes smoked by 5 per day increased the risk of stroke by 12% [5,6]. Larsson et al., (2019) indicated that even smoking just one cigarette a day was associated with an approximately 25 to 30% increased risk of stroke [7]. Depending on the number of cigarettes smoked, the risk of atrial fibrillation is also increased [8]. Other authors suggest that secondhand smoke also increases the risk of stroke [9].

Tobacco smoke promotes the development of atherosclerosis, which secondarily leads to local obstruction and embolism, but is also associated with sleep apnea, migraine, and gout. In addition, smoking is associated with other risk factors for stroke, such as diabetes, high blood pressure, and increased resting heart rate [6,10]. Platelets, which are among the components of the blood–vascular axis, play a key role in the development of focal cerebral ischemia. Activated platelets initiate hemostatic plug formation and provide a scaffold for the activation of coagulation [11]. They are also responsible for maintaining “inflammatory homeostasis”, which refers to the integrity of the vessels at the site of inflammation [12]. The inflammatory response to smoking plays a significant role in the development of atherosclerotic lesions, but the pathomechanisms that accompany stroke are not entirely clear. Fatty acids in the form of synthesized arachidonic and linoleic acid derivatives are released from platelet cell membranes. Eicosanoids play an important role in the regulation of cardiovascular disease processes. Drugs that indirectly modulate these molecules are used to treat atherosclerotic thrombosis, including COX-1/COX-2 inhibitors [13]. Cerebral ischemia breaks the dynamic balance between pro-inflammatory and anti-inflammatory responses. Increased inflammation plays a significant role in the progression of ischemic stroke. Inflammation is characterized by an increase in the levels of markers such as C-reactive protein (CRP), interleukin (IL)-6 and IL-1, and tumor necrosis factor (TNF), especially in smokers. Smokers also have higher leukocyte counts and increased recirculation of leukocytes, which is the basis of atherosclerotic lesions, but also intensifies the incidence of stroke. Inhibiting inflammatory responses may reduce brain damage and improve neurological outcomes [14]; therefore, understanding these mechanisms is of great practical importance, as it allows the selection of an appropriate treatment regimen and drugs for a given patient. It has been shown that inhibiting inflammatory responses can reduce brain damage and improve neurological outcomes [14].

The literature discusses the influence of gender on the incidence of stroke and related treatment methods. The available literature suggests different pathomechanisms of stroke and clinical pictures in men and women [15,16]. Data on the differences between men and women are complex, and the differences depend on age, among other factors. Up to 45 years of age, both sexes have similar mortality rates from stroke, but at 45 and older, the rate for women affected by stroke increases. Women are believed to have a higher risk of stroke throughout their lives, and they also have recurrent strokes. This is probably associated with a decrease in the neuroprotective effect of estrogen post-menopause. Men are at higher risk of ischemic stroke and women are more likely to suffer from subarachnoid hemorrhage [17]. There are also differences in stroke sub-types depending on gender. Women are more likely to have cardioembolic strokes and men are more likely to have lacunar and atherosclerotic strokes [18]. Other authors have found that stroke tends to be more severe in women, with a 1-month mortality of 24.7%, compared with 19.7% in men [19]. With regard to vascular risk factors, female gender is an independent risk factor for thromboembolism [20,21]. Women also have more severe neurological deficits [19].

The relationships between gender and stroke and smoking and stroke are documented in the literature, but mainly from observational studies. The exact gender-specific mechanism by which cigarette smoke induces stroke inflammation is still unclear. The aim of the presented study was to determine the impact of cigarette smoking on inflammatory markers such as eicosanoids. To the best of our knowledge, there is no literature on the effects of both gender and smoking on lipid markers of inflammation after ischemic stroke.

## 2. Materials and Methods

### 2.1. Characteristics of the Study Group

The study included 73 Caucasian patients who were hospitalized after ischemic stroke in the neurology department of a county hospital in the western part of Poland. Informed consent was obtained from all subjects. The diagnosis of ischemic stroke was made on the basis of clinical symptoms and the results of additional tests. The exclusion criteria were as follows: intracranial hemorrhage, impaired speech or consciousness, and the presence of an active infection (body temperature above 37.4 °C). Patients with typically inflammatory diseases such as autoimmune rheumatic diseases, colitis, or cancer were not recruited for the study. Standard treatment of ischemic stroke was used in accordance with the guidelines of the Expert Group of the Vascular Section of Brain Disease of the Polish Neurological Society [22] and the guidelines for the prevention of stroke in patients with stroke and transient ischemic attack by the American Heart Association/American Stroke Association [23].

Patients were divided according to smoking status as well as gender and smoking status. In the first division, there were 46 patients in the group of non-smokers and 27 in the group of smokers; 24 women and 22 men were classified as non-smokers, while 16 women and 11 men were smokers. Group characteristics are presented in Appendix A.

### 2.2. Isolation and Determination of Eicosanoids

Fasting blood samples were collected in EDTA-containing tubes on the seventh day of hospitalization. After centrifugation, the supernatant was further stored at −80 °C. Isolation of eicosanoids from plasma was carried out with the use of solid-phase extraction using RP-18 SPE columns (Agilent Technologies, Cheadle, UK). Then, 1 mL of chilled acetonitrile to precipitate the protein and 50 μL of internal standard PGB2 (1 μg/mL) were added to 0.5 mL of plasma. After 15 min incubation at −20 °C, the samples were centrifuged at 10,000 rpm for 10 min (Eppendorf 5804R centrifuge). The supernatants were transferred to new collection tubes containing 4.5 mL of 1 mM HCl. Each sample was adjusted to pH 3 by adding 30–50 µL of 1 M HCl. The columns were activated with successive washes of 3 mL of 100% acetonitrile and 3 mL of 20% acetonitrile in water. After the impurities were washed off with 20% acetonitrile in water, 1.5 mL of a mixture of methanol and ethyl acetate (1/1 *v*/*v*) was eicosanoid eluted. The samples were then dried under vacuum and dissolved in 100 μL of 60% methanol in water with 0.1% acetic acid immediately before HPLC analysis.

The following compounds were analyzed: 5(S),6(R)-Lipoxin A4 (cat. no. 20110), 5(S),6(R),15(R)-Lipoxin A4, (cat. no. 20110), 5 HETE, (cat. no. 34230), 5-oxo ETE (cat. no. 34250), 12 HETE (cat. no. 34570), 15 HETE, (cat. no. 34720), 16(R)/16(S) HETE (cat. no. 10004385/10004384), 9 HODE (cat. no. 38410), 13 HODE, (cat. no. 38610), 18 HEPE (cat. no. 3284), 17 HDHA (cat. no. 3365), 10(S)17(R) DiDHA (Protectin DX; cat. no. 10008128), Maresine 1 (cat. no. 10878), Leukotriene B4 (cat. no. 20110), Prostaglandin E2 (cat. no. 14010), Prostaglandin B2 (cat. no. 11210), Resolvin D1 (cat. no. 10012554), Resolvin E1 (cat. no. 10007848), and TXB2 (cat. no. 10007237). Separation of the analyzed compounds was carried out using an Agilent 1260 liquid chromatograph on a Thermo Scientific Hypersil BDS C18 column (2.4 μm, 100 × 4.6 mm; cat no. 28102–154630) at 20 °C.

Mobile phase A was methanol/water/acetic acid (50/50/0.1, *v*/*v*/*v*) and solvent B was methanol/water/acetic acid (100/0/0.1, *v*/*v/v*). Mobile phase B buffer content was 30% at 0 to 2 min of separation, increased linearly to 80% at 33 min, was 98% between 33.1 and 37.5 min, and 30% between 40.3 and 45 min. The flow rate was 1.0 mL/min.

The DAD detector monitored peaks by adsorption at 235 nm for 16-HEPE, 17-HDHA, 9-HODE, 13-HODE, 5-HETE, 12-HETE and 15-HETE, at 280 nm for PGB2 (prostaglandin B2, internal standard) and 5oxoETE, Rev E1, Protectin DX, Maresine1, Leukotriene B4 at 210 nm for PGE2, 16-HETE and TXB2 and at 302 nm for 5(S),6(R)-lipoxin A4, 5(S),6(R)), 15(R)-lipoxin A4 and RevD1. The quantification of analytes was determined against the calibration curve, determined separately for each compound and automatically corrected by the value of the internal standard. Results analysis was performed using ChemStation software (Agilent Technologies, Cheadle, UK) [24].

### 2.3. Statistical Analysis

Statistical analysis was performed with Statistica 12 (Statsoft, Kraków, Poland). For comparison most of variables between groups, the parametric tests were used after confirmation of the normal distribution by Shapiro-Wilk test. Non-parametric Mann-Whitney test was used for comparisons between two groups, ischemic stroke and control groups, in which *p* < 0.05 was considered statistically significant. The post hoc power analysis was performed with using the G power software (Dusseldorf, Germany). The correlograms was done with using OriginPro2021.

The effect size analysis, i.e., the Cohen coefficient, was performed for statistically significant data and performed using https://lbecker.uccs.edu/, accessed on 11 November 2022. This coefficient determines that if the difference between the means of the two groups is less than 0.2 standard deviation, the difference is not significant, even if it is statistically significant. A factor of less than 0.5 is “medium” effect size, and 0.8 is “large” effect size. The application of this coefficient allowed for additional verification of the obtained results.

## 3. Results

### 3.1. Biochemical Analysis of Eicosanoids

In the first stage of results analysis, post-stroke patients were divided into smokers and non-smokers, regardless of gender. Total non-smokers vs. total smokers differed only in the level of linolenic acid (LA) derivative 9 HODE (*p* < 0.05), with a higher concentration in TNS. Cohen’s coefficient was, however, at the level of 0.15, indicating the insignificance of the difference (Table 1).

Therefore, in the next step, post-stroke patients were divided by gender and smoking status. After dividing the results by sex, it turned out that the 9 HODE level was higher in female smokers than non-smokers (*p* ≤ 0.01, Cohen’s d = 0.72), while there was no statistically significant difference in the group of men. Similarly, the level of arachidonic acid metabolite LTX A4 (*p* ≤ 0.05, Cohen’s d = 0.53) showed a statistically significant difference only in the group of women. In this group, we also observed decreased 15 HETE levels in female smokers, but it was statistically insignificant (*p* ≤ 0.08) but Cohen’s d = 0.72), so it was interpreted as a trend (Table 2). On the other hand, the level of 15 HETE was statistically significantly higher in the group of female non-smokers compared to male non-smokers (*p* ≤ 0.03, Cohen’s d = 0.57) (Table 3).

In the group of men, statistically significant differences were observed in relation to two compounds, TXB2 and NPD1. The levels of TXB2 were almost two times higher (*p* ≤ 0.01, Cohen’s d = 1.12) in men who smoked than men who did not smoke, and in this group we also observed increased levels of NPD1 compared to male non-smokers (at the limit of significance, *p* ≤ 0.06 but Cohen’s d = 0.53) (Table 2). Among male non-smokers, in addition to the difference in 15 HETE levels described above, were also observed elevated TXB2 levels (*p* ≤ 0.01, Cohen’s d = 1.10). Interestingly, the statistical analysis did not show any differences between men and women who smoked (Table 3).

### 3.2. Eicosanoid Correlation Analysis of Individual Patient Groups

Statistically significant relationships are presented in the description of correlations. Due to their number, they have been described in relation to specific groups of patients. Spearman’s correlation analysis showed the following relationships (*p* ≤ 0.001).

#### 3.2.1. Female Non-Smokers

The main metabolites of linolenic acid (LA), 9 and 13 HODE, in addition to correlating between themselves (r = 0.61), were positively correlated with Maresin R1 (MaR1) (r = 0.51 and r = 0.70, respectively) and 15 HETE, the main metabolite of the 15-LOX enzyme of the AA arachidonic acid pathway (r = 0.52 and r = 0.61, respectively). The increase in 13 HODE was also associated with an increase in the remaining arachidonic acid metabolites, 12 HETE (r = 0.67) and 5 HETE (r = 0.49), and metabolites of the eicosapentaenoic acid (EPA) pathway, 18-HEPE (r = 0.54) and Resolvin E1.

NPD1, a metabolite of doxahexaenoic acid (DHA), showed a number of correlations with the previously described 13 HODE (r = 0.48), with Resolvin E1 (r = 0.52) and 18 HEPE (r = 0.58) (EPA pathway), Maresin 1 (r = 0.51) (DHA pathway), 5 HETE (r = 0.46), 12 HETE (r = 0.68), 15 HETE (r = 0.41), and TXB2 (r = 0.71) (AA pathway). The levels of another DHA metabolite, Maresin R1, changed with the change of Leukotriene B4 (r = 0.52) (the only correlation of this relationship), 13 HODE (r = 0.70), 9 HODE (r = 0.51), 18 HEPE (r = 0.61), NPD1 (r = 0.51), 5 HETE (r = 0.70), 12 HETE (r = 0.52), and 15 HETE (r = 0.64).

The compound 18 HEPE was among those with the most numerous correlations. Its increase was related to an increased level of Resolvin E1 (r = 0.57), which proves the activity of 15 LOX enzyme, for which 18 HEPE is a substrate. Correlations were also observed between Maresin 1 (r = 0.61), 13 HODE (0.54), 15 HETE (r = 0.41), 12 HETE (r = 0.55), 5 HETE (0.61), TXB2 (r = 0.64) and NPD1 (0.58). Resolvin E1 was additionally correlated with 13 HODE (r = 0.44), NPD1 (r = 0.52), and 12 HETE (r = 0.53).

Thromboxane B2, a metabolite of the arachidonic acid pathway with the participation of 1/2 COX, was positively correlated with 5 and 12 HETE (r = 0.58 and 0.46, respectively) and NPD1 (r = 0.71) and 18 HEPE (r = 0.64) The 5 HETE level, in addition to the above-mentioned correlations, increased with increased 12 HETE (r = 0.47) and 15 HETE (r = 0.53). For 5-oxo ETE, LXA4, PGE2, PD1, Resolvin D1, and 16 HETE, no correlation was observed in the analyzed group (Figure 1a).

#### 3.2.2. Female Smokers

Less statistically significant correlations were found in the group of smoking women compared to non-smokers. Several eicosanoids showed no relationship with each other: LTB4, 12 HETE, PD1, Resolvin D1, LXA4 and 16 HETE (the latter two compounds were below the limit of quantification). As in non-smoking women, mutual interactions between 9 and 13 HODE (r = 0.82) and a positive correlation with 15 HETE (r = 0.74 and 0.66, respectively) were observed. 13 HODE also positively correlated with 5-oxo ETE (r = 0.61). 5 oxo ETE is the result of the transformation of 5 HETE (r = 0.54). Its level was related to TXB2 (r = 0.54) and 15 HETE (r = 0.67). TXB2 showed no other correlations, while 15 HETE also correlated with 5 HETE (0.64) and NPD1 (r = 0.54).

NPD1 also showed an additional correlation with 18 HEPE (r = 0.53). 9 HODE was positively associated with Resolvin E1 (r = 0.53). The only negative correlation was found between the main arachidonic acid metabolite (enzyme product COX-1/2) Prostaglandin E2, which was negatively correlated with Maresin R1 (r = −0.64) (Figure 1b).

#### 3.2.3. Male Non-Smokers

In this group, as in the group of women who did not smoke, many mutual correlations were observed. The most numerous correlations were observed for eicosanoids derived from linolenic acid (LA) (9 and 13 HODE) and arachidonic acid (AA) (15 HETE, 5 HETE, 5-oxo ETE, LTB4, and DHA-Maresin R1 docosoeicosanoid).

Correlations were found between 9 and 13 HODE (r = 0.95) and 15 HETE (r = 0.74 and 0.78, respectively), 5-oxo ETE (r = 0.45 and 0.49), LTB4 (r = 0.87 and 0.87), Maresin 1 (r = 0.53 and 0.50), 18 HEPE (r = 0.59 and 0.62), and RvD1 (r = 0.45 and 0.54).

Furthermore, 13 HODE was also correlated with NPD1 (r = 0.46), and 15 HETE was positively associated with 18 HEPE (r = 0.48), NPD1 (r = 0.66), Resolvin D1 (r = 0.49), Maresin 1 (r = 0.50), Leukotriene B4 (r = 0.65), and 5-oxo ETE (r = 0.65). In addition, 5-oxo ETE was also correlated with PDN1 (r = 0.72), 18 HEPE (r = 70), Leukotriene B4 (r = 0.48), Maresin 1 (r = 0.65), and Leukotriene B4 (r = 0.48). Leukotriene B4 was also correlated with Prostaglandin E2 (r = 0.53), Maresin 1 (r = 0.53), and 5 HETE (r = 0.50). The fourth PGE2 correlation was the relationship with Maresin 1 (r = 0.48). The PD1 level was interrelated with the NPD1 level (r = 0.45). Along with Thromboxane B2, 5 HETE changed in the same trend (r = 0.63). We found no correlation for three metabolites: LXA4, Resolvin E1, and 16 HETE, the latter of which was below the quantification limit (Figure 1c).

#### 3.2.4. Male Non-Smokers

As in the group of smoking women, less correlation was observed in the group of smoking men than in the group of non-smokers. In contrast to non-smokers, no correlation was found for PGE2 and resolvin D1 in this group. However, new correlations were observed for LXA4 and Resolvin E1.

LXA4 levels increased with increasing 15 HETE (r = 0.62) and NPD1 (r = 0.61), and Resolvine E1 levels were associated with 12HETE (r = 0.73), 5oxoETE (r = 0.75), and Maresine 1 (r = 0.73). The most numerous correlations, as in the group of non-smokers, were observed for Mar1, LTB4, 5HETE, 5oxoETE, 15HETE. Maresine 1 grew together with Leucotriene B4 (r = 0.86), 12HETE (r = 0.83), 5HETE (r = 0.78), 5oxoETE (r = 0.75), TXB2 (r = 0.74), and the already mentioned Resolvine E1.

LTB4 level was associated with Maresin 1 (r = 0.86), NPD1 (r = 0.65), 15 HETE (r = 0.78), 12 HETE (r = 0.72), 5 HETE (r = 0.62), 5-oxo ETE (r = 0.76), and TXB2 (r = 0.78). Among arachidonic acid derivatives, 5 oxoHETE- Resolvine E1 (r = 0.75), Maresine 1 (r = 0.75), Leucotriene B4 (r = 0.76), 15 HETE (r = 0.66), NPD1 (r = 0.64) and 12 HETE (0.84). 5 HETE which is the precursor of 5oxoETE, did not correlate with each other, and its level depended on Marsien 1 (r = 0.78), Leucotriene B4 (r = 0.62), and TXB2 (r = 0.65). 15 HETE correlated positively with LTX4 (r = 0.62), Leucotriene B4 (r = 0.78), 5oxoETE (r = 0.66) and TXB2 (r = 0.64).

In contrast to the group of non-smokers, more numerous correlations were observed for TXB2 and 12 HETE, and less numerous for 9 and 13 HODE, and 18 HEPE. The level of TXB2 was related to the level of Maresine 1 (r = 0.74), Leucotriene B4 (r = 0.78), 15 HETE (r = 0.64), and 5 HETE (r = 0.65). In the same trend as 12 HETE, the level of Resolvine E1 9r = 0.73), Maresine 1 (r = 0.83), Leucotriene B4 (r = 0.72), NPD1 (r = 0.63), and 5oxoETE (r = 0.84) changed.

HODE levels 9 and 13 were associated with NPD1 (r = 0.75, r = 0.75, respectively) and, as in all previously described groups, these relationships correlated with each other (r = 0.99). For 18 HEPE, only one relationship with LTB4 was shown (r = 0.65) (Figure 1d).

## 4. Discussion

All the HODE derivatives are particularly atherogenic, and their activity is modulated by lymphocytes [25]. Interpreting mediator levels without taking into account fatty acid levels is very difficult and should be done with care. This is because it is not possible to capture the entire course of the pro-inflammatory cascade. The course is continuous, as is the use of synthesized mediators. Therefore, low levels of mediators may indicate a lack of their synthesis, as well as an inefficient system and the use of mediators in an inflammatory reaction. Only changing fatty acid levels would likely allow correct interpretation. The levels of fatty acids in the study group were analyzed, however, only the results for pro- and anti-inflammatory mediators were presented in this paper.

### 4.1. Inflammatory Response in Male and Female Non-Smokers

In previous research by our team, analyzing the level of eicosanoids in people after ischemic stroke, no differences were detected between women and men [24] except for a difference in thromboxane levels (data not published). Women had higher TXB2 levels than men (0.09 vs. 0.06; *p* ≤ 0.001). Regarding our current analysis, it should be considered that the results are due to an increased level of TXB2 in female non-smokers (Table 3). In the analysis presented in this study, the increased level of 9 HODE in the group of non-smokers (Table 1) is also attributed to non-smoker women.

Elevated thromboxane in non-smoking women may be associated with platelet activation during stroke, which leads to the activation of platelet integrin GPIIbIIIa, the release of granularity, including the ADP feedback agonist, and the synthesis of thromboxane B 2 (TXB 2), which promotes the recruitment of circulating platelets in the blood. TXB 2 was also positively correlated with other anti-inflammatory lipid mediators, such as NPD1 and inflammatory 12 HETE and 5 HETE. It can be conservatively concluded that women who are not exposed to additional environmental factors such as smoking should have a better prognosis.

By analyzing the correlations, it has been observed that mediators (eicosanoids) that participate in the inflammatory response can be identified, regardless of gender; however, this response is differentiated. The following compounds have been included: 9 and 13 HODE, 5 HETE, LTB4, 12 and 15 HETE, TXB2, 18 HEPE, NPD1, and Maresin 1. Some mediators were found to be characteristic of men and others of women (Appendix A). Resolvin E1 was involved in the inflammatory response and its suppression in women, and 5-oxo ETE, PGE2, and Resolvin D1 in men. Statistically significant elevated levels of the following mediators were observed: TXB2, 15 HETE and NPD1 (*p* ≤ 0.07) in non-smokers women compared with non-smokers men (Table 3). 

### 4.2. Inflammatory Response in Male and Female Smokers

Interestingly, there were no statistically significant differences in the level of the analyzed eicosanoids in this group. However, the correlation analysis determined which inflammatory and anti-inflammatory mediators are involved in the post-stroke response: 9 and 13 HODE, responsible for oxidative stress and atherogenic changes; 5 HETE and 5-oxo ETE; 15 HETE, TXB2, 18 HEPE and RvE1; and NPD1 and Maresin 1. Meanwhile, the gender-specific mediators are PGE2 in women who smoked, and LTB4, LXA4, and 12 HETE in men (Appendix A). The involvement of more pro-inflammatory than anti-inflammatory mediators in men may partially explain why men have a worse prognosis in the first post-stroke period than women.

The severity of the inflammatory response, inferred from the number of relationships between compounds, shows that smoking tobacco had a stronger effect on the inhibition of inflammatory response pathways in women than in men. However, it seems that the post-stroke inflammatory response in smokers may be more complicated and may result from the faster use of substrates for lipid mediators, such as fatty acids (arachidonic acid, linoleic acid, EPA, and DHA). However, analyzing the level of these relationships was not the focus of this work. The low-grade inflammation that leads to stroke is long-lasting. It is associated with body dysfunctions such as disturbances in carbohydrate and lipid metabolism, leading to the onset of diabetes and atherosclerosis. The influence of these factors on the obtained results cannot be excluded; therefore, the presence of these diseases may be considered as a certain limitation of the research. However, smoking cigarettes is one of the important factors that will increase inflammation. This is evidenced by the fact that smokers experience stroke events several years earlier than non-smokers [7].

### 4.3. Inflammatory Response in Female Smokers and Non-Smokers

Lipid mediators formed from arachidonic acid have both protective and inflammatory effects. Higher levels of LTX4, 9 HODE, and 15 HETE were observed in women who did not smoke compared with women who did (Table 2). ARA serves as a substrate for 5/12/15 LOX for the production of leukotrienes and lipoxins. Leukotrienes are cytotoxic and inflammatory, while lipoxins A4 and B4 (synthesized by 5 and 15 lipoxygenases) [26] have anti-inflammatory and immunomodulatory properties [27]. In a rat ischemic stroke model, LTX4 was observed to ameliorate the course of stroke by inhibiting the matrix metallopeptidase 9 (MMP-9) pathway, dependent on nuclear factor kappa B (NF-κB) and regulating oxidative stress [26,28]. Interestingly, LTX4 did not show any correlation with any of the analyzed eicosanoids in the analyzed groups of women, including 15 HETE, which is a precursor of LTX4 (Figure 2). 15 HETE plays an important role in post-ischemic cerebrovascular contractility. In addition, studies have shown that increased expression of 15-lipoxygenase (15-LO), which converts arachidonic acid to 15-hydroxyeicosatetrienic acid (15 HETE), is associated with ischemia and hypoxia in the cerebral vascular endothelium. It is a time-dependent process and is most pronounced in the later stages of a stroke [29]. Wang et al. [29] also demonstrated a key role of 15-LOX/15 HETE in behavior and angiogenesis after stroke and functional regeneration. In experimental mice, the infarct volume gradually decreased after stroke as 15-LOX expression increased. The authors found that 15-LOX/15 HETE promotes angiogenesis and poststroke neuronal regeneration in mice, and 15 HETE stimulates the proliferation and migration of brain microvascular endothelial cells (BMVECs) through PI3K/Akt pathway signaling. It has also been proven that hypoxia stimulates the proliferation of pulmonary arteries, which is a key element of vascular angiogenesis [30]. The compound 15 HETE was among those that showed the most correlations in the group of female non-smokers. On the other hand, the reduced level of 15 HETE in the group of female smokers was associated with worse cognitive function, already observed by our team [31], together with 5 HETE, 12 HETE, and Maresin 1.

Derivatives of the linoleic acid pathway, 9 and 13 HODE, were observed in all of the analyzed groups, but only this comparison showed a statistically significantly higher level of 9 HODE in women who did not smoke. Their presence is associated with oxidative stress in fluids and biological systems, as well as inflammation, stroke, atherosclerosis, and metabolic syndrome [31]. They are even considered to be more specific markers of inflammation than the typically used isoprostanes [32]. In studies by other scientists, increased levels of these derivatives were also observed in patients after stroke and vascular dementia [32]. Of note, 9 and 13 HODE have opposite effects: pro-inflammatory and anti-inflammatory, respectively [33]; although, they are also highly atherogenic, and their activity is modulated by Th1 and Th2 lymphocytes with the participation of cytokines and chemokines [25]. Further, 9 HODE is the most potent ligand involved in the pathogenesis of atherosclerosis, via GPR132; but on the other hand, 13 HODE affects the adhesion of platelets to the endothelial wall [34]. Probably, the induction of inflammation in non-smokers is an element of more intensive post-stroke regeneration, resulting from the involvement of a number of pro-resolving and anti-inflammatory mediators, in contrast to smoking women.

Another metabolite of major importance in stroke is 12 HETE. This derivative did not differentiate the group of women, but differences in correlations were observed. There was no dynamic relationship between 12 and 15 HETE in any group. No correlation was found between this mediator and women who smoked, while among women who did not smoke the levels of 12 HETE varied with changes in arachidonic acid metabolites (5 HETE and TXB2), promoter metabolites of the EPA pathway (18 HEPE and Resolvin E1) and DHA pro-binding metabolites (NPD1 and Maresin R1). The synthesis of 12 HETE is associated with the body’s response to oxidative stress and AA oxidation, and is especially intensified in smokers [13]. However, this dependence has not been observed in women who smoke. In this group, the anti-inflammatory Maresin 1 was negatively correlated with prostaglandin E2. PGE2 is a major pro-inflammatory mediator that has neurotoxic properties, but may also inhibit platelet aggregation [35]. A role for PGE2 in the development of dementia, vascular dementia and Alzheimer’s disease has been reported. Fewer correlations were found among smokers than non-smokers. We assume that there may be various mechanisms for this phenomenon, related, on the one hand, to the inhibition of the activity of enzymes of analyzed biochemical pathways 5-, 15-, and 12-LOX by tobacco smoke, and on the other hand, to the failure of the protective system, because it was already intensified before the occurrence of stroke due to oxidative stress. Obviously, both hypotheses require further research.

### 4.4. Inflammatory Response in Male Smokers and Non-Smokers

Concentrations of thromboxane B2 and NPD1 were statistically significantly higher in smoking men than in non-smokers (Table 2). Thromboxane B2 (TXB 2) and the antagonistic prostaglandin I 2 (PGI 2) are metabolites of arachidonic acid and play a key role in the pathogenesis of ischemic stroke. Overproduction of TXB 2 is one of the key factors in thrombosis, stroke, and heart disease. Excessive production of pro-inflammatory PGE 2 and pro-thrombotic TXB 2, with reduced PGI 2 anticoagulant biosynthesis, has been shown to increase the risk of stroke and myocardial infarction [36]. Interestingly, Kashiwagi et al. [37] proved that cigarette smoke extract (CSE) inhibits the production of TXA 2 in platelets along the COX-1 pathway, depending on the concentration, by binding to the COX-1–PGG 2 complex, which is a precursor to TXA. However, in the central nervous system, COX-1 is constitutively expressed in neurons, astrocytes, and microglial cells. It is not known if this pathway is also inhibited by CSE. On the other hand, CSE also enhances the activity of TX synthase in a concentration-dependent manner. Higher levels of TXB 2 in male smokers may also be associated with increased activation of COX-2 cyclooxygenase, which is localized in macrophages, leukocytes, and fibroblasts. Activation of this pathway occurs in pathological conditions [38] as a response to increased inflammation [39]. It has been shown that vascular endothelial dysfunction is also characterized by increased production of prostanoids (including TXB 2), which facilitates the penetration of macrophages into the vessel wall [40].

NPD1 is a metabolite of docosahexaenoic acid (DHA) that serves as a precursor to 17 (S)-resolvin [41,42]. It has anti-inflammatory and oxidative stress-extinguishing properties, and the ability to repair tissues and reduce neutrophil recruitment. Neuroprotectin D1 has also been shown to confer protection after ischemic brain injury in a mouse stroke model by blocking mitochondria-mediated cell death pathways [43]. We suppose that higher levels of this eicosanoid may be dictated by a tendency to compensate for increased oxidative stress. Moreover, NPD1 was positively correlated with LTX A4 lipoxin, which can be synthesized from 15 HETE (Figure 3) [44]. 15 HETE also showed a positive correlation with LTX A4. These correlations may be explained by the fact that lipoxin synthesis takes place in platelets and/or leukocytes. In the case of lipoxin synthesis from peripheral platelets, 15-lipoxygenase (15-LOX) introduces oxygen into AA, which leads to the production of 15-hydroxyeicosatetraenoic acid (15 HETE), which is converted to lipoxins by 5-lipoxygenase (5-LOX) [39,45]. Lipoxins are anti-inflammatory molecules that promote inflammation relief and tissue regeneration. They are considered to be among the most important lipid derivatives involved in ischemic stroke [46]. Their role is related to a reduction in the amount of reactive oxygen species and the production of pro-inflammatory cytokines and chemokines, as well as blocking adhesion and transmigration through the endothelium and inhibiting the synthesis of pro-inflammatory leukotriene B4 [45,47,48].

In male smokers, a positive correlation was also observed between NPD1, 12 HETE and 5-oxy ETE. Contrary to 15 HETE, 12 HETE is a molecule that stimulates chemotaxis and chemokinesis in leukocytes. It is formed by the oxidation of AA by 12-LOX enzyme, while reducing the synthesis of anti-inflammatory EET derivatives, which in turn is associated with cigarette-induced oxidative stress [13]. The body’s response to elevated levels of 12 HETE is increasing Maresin 1 and Resolvin E1 (positive correlations). Maresin 1 was also positively correlated with 5-oxo ETE, 5 HETE, and TXB 2, whereas Resolvin D1 showed no correlation. Maresin 1, a derivative of DHA, is a pro-inflammatory and anti-inflammatory molecule. Its protective role against brain tissue has been described in a mouse model of cerebral ischemia [49].

When analyzing the correlations of lipid mediators involved in the post-stroke response, common features were also observed for all groups. Mediators that were involved in post-stroke inflammation, irrespective of smoking and gender, were: TXB2, 18 HEPE, 9 and 13 HODE, 15 HETE and 5 HETE, and Maresin 1 (Appendix A). Our research highlights the effect of smoking on inflammation in both genders mediated by lipid mediators, which makes smoking cessation undeniable.

## 5. Conclusions

The above results indicate that the inflammation that occurs in acute ischemic stroke differs in both severity and in the pro-inflammatory pathways activated, depending on gender and smoking status. In general, the triggered response was more intense in non-smokers, as indicated by a much greater volume of cross-correlations than in smokers. This may be related to the inhibition of fatty acid conversion pathways by tobacco smoke, but also to the “breakdown” of the inflammatory response system, one of the body’s defense mechanisms, which becomes ineffective at this point. Therefore, in smokers, tobacco smoke both increases and accelerates oxidative stress before a stroke event. Arachidonic acid derivatives such as leukotriene B4 and 5-oxo ETE were also found to be involved in male non-smokers. Leukotriene B4 can be considered gender-related, as higher levels were not observed in either group of women. Its many correlations in men may be associated with the observed worse prognosis for men than for women, because LTB4 induces an acute inflammatory response and exacerbates brain damage.

Additionally, an EPA derivative (Resolvin E1) was involved in female non-smokers, and two DHA derivatives (Resolvin D1 and Protectin D1) in male non-smokers. In practice, it can be concluded that properly selected supplementation for prevention and during convalescence can strengthen the anti-inflammatory response pathways.

## 6. Study Limitations

The limitation of the study concerns the population (only the Caucasian race) and the limited number of patients in individual groups, however, the specificity of the study group did not allow for the collection of a larger number of patients. To additionally verify the obtained results, the Cohen coefficient was used. Comorbidities such as diabetes and atherosclerosis were also taken into account. Their impact on the study cannot be completely ruled out, but this requires more detailed analysis, which was not the subject of this experiment.

## Figures and Tables

**Figure 1 biomedicines-11-00092-f001:**
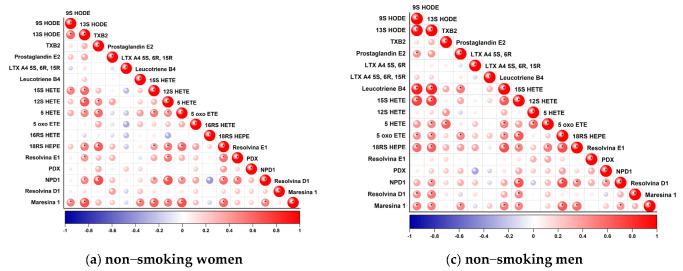
Correlation plot showing correlations between particular eicosanoids in studied groups (**a–d**). Red indicates a positive association; blue indicates an inverse association. Higher color intensity and larger circles indicate stronger associations; lower color intensity and smaller circles indicate weaker associations. Asterisk indicates statistically significant correlations.

**Figure 2 biomedicines-11-00092-f002:**
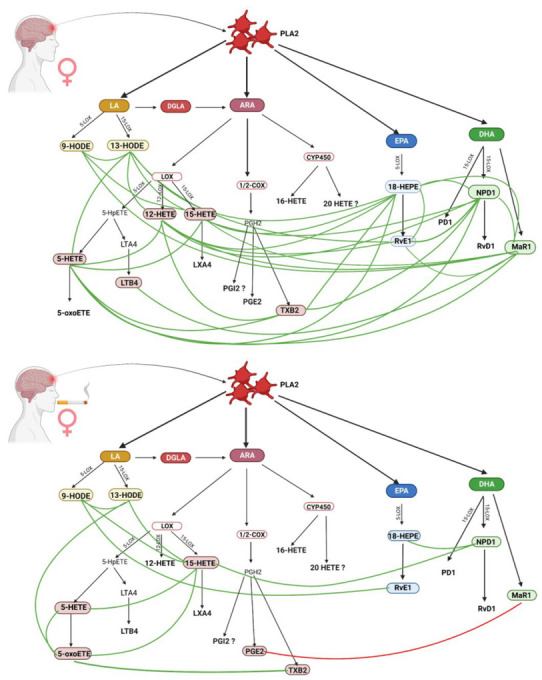
Comparison of inflammatory response in female non-smokers and female smokers. (Created with BioRender.com https://app.biorender.com/; accessed on 30 September 2022).

**Figure 3 biomedicines-11-00092-f003:**
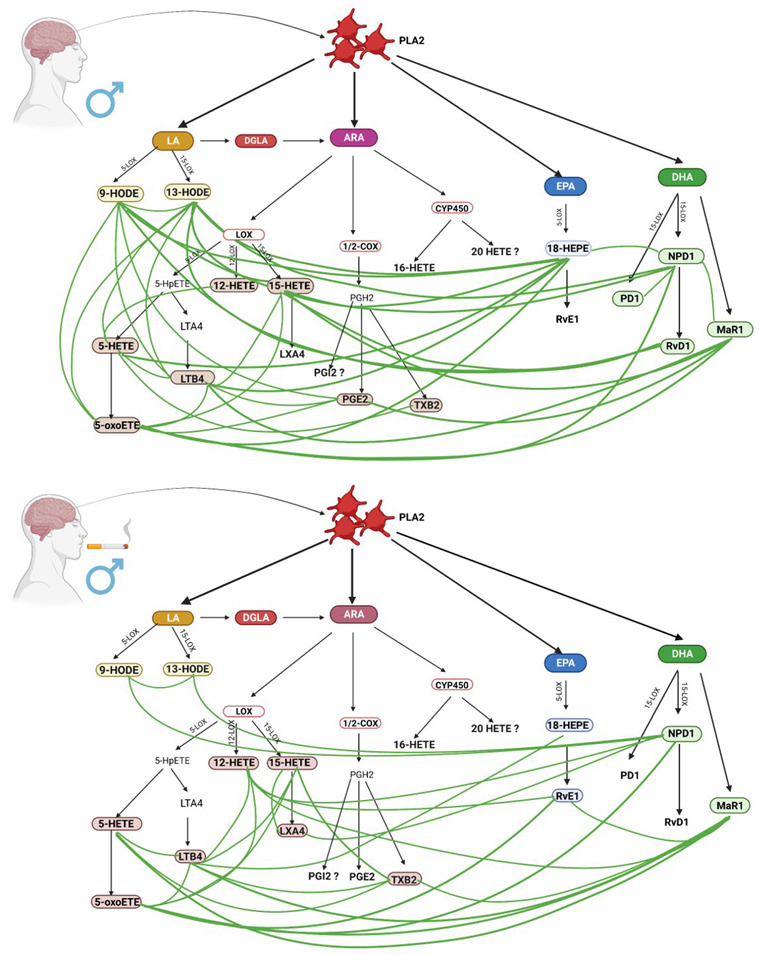
Comparison of inflammatory response in male non-smokers and male smokers. (Created with BioRender.com https://app.biorender.com/; accessed on 30 September 2022).

**Table 1 biomedicines-11-00092-t001:** Eicosanoid levels in groups of smokers and non-smokers, excluding gender.

Eicosanoids (ng/mL)	Total Non-Smokers	Total Smokers	*p* Value	Cohen’s d
	Avg ± SD	Avg ± SD		
	n = 46	n = 27		
LA metabolites				
9 HODE	0.035 ± 0.021	0.030 ± 0.040	0.05 *	0.15
13 HODE	0.033 ± 0.022	0.033 ± 0.040	0.20	-
AA metabolites				
TXB2	0.081 ± 0.066	0.085 ± 0.052	0.43	-
Prostaglandin E2	3.653 ± 5.066	2.488 ± 1.916	1.00	-
LTX A4 5S, 6R	0.028 ± 0.187	0.000 ± 0.000	0.88	-
LTX A4 5S, 6R, 15R	0.022 ± 0.037	0.018 ± 0.039	0.38	-
Leukotriene B4	0.027 ± 0.014	0.026 ± 0.014	0.73	-
15 HETE	0.292 ± 0.212	0.291 ± 0.198	0.70	-
12 HETE	1.799 ± 1.154	1.671 ± 1.016	0.73	-
5 HETE	0.026 ± 0.015	0.024 ± 0.008	0.70	-
5-oxo ETE	0.190 ± 0.111	0.175 ± 0.082	0.70	-
16 HETE	0.016 ± 0.086	0.000 ± 0.000	0.76	-
EPA metabolites				
18 HEPE	0.107 ± 0.030	0.113 ± 0.048	0.92	-
Resolvin E1	0.048 ± 0.068	0.072 ± 0.120	0.38	-
DHA metabolites				
Protectin D1 (PDX)	0.048 ± 0.073	0.044 ± 0.048	0.96	-
NPD1	0.120 ± 0.092	0.119 ± 0.067	0.68	-
Resolvin D1	0.204 ± 0.297	0.124 ± 0.160	0.40	-
Maresin 1	0.031 ± 0.015	0.031 ± 0.018	0.91	-

LTX, lipoxin; HETE, hydroxyeicosatetraenoic acid; HODE, hydroxyoctadecadienoic acid, Avg—mean, SD—standard deviation, * statistically significant.

**Table 2 biomedicines-11-00092-t002:** Eicosanoid levels in smokers and non-smokers, including gender.

Eicosanoids (ng/mL)	Women		*p* Value	Cohen’s d	Men		*p*-Value	Cohen’s d
	Non-Smokers	Smokers			Non-Smokers	Smokers		
	Avg ± SD n = 24	Avg ± SD n = 16			Avg ± SD n = 22	Avg ± SD n = 11		
LA metabolites								
9 HODE	0.036 ± 0.019	0.023 ± 0.017	0.01 *	0.72	0.032 ± 0.024	0.041 ± 0.054	0.95	
13 HODE	0.032 ± 0.020	0.027 ± 0.02	0.11	-	0.033 ± 0.024	0.041 ± 0.057	0.80	
AA metabolites								
TXB2	0.105 ± 0.072	0.072 ± 0.047	0.21	-	0.046 ± 0.023	0.086 ± 0.045	0.01 *	1.12
Prostaglandin E2	4.101 ± 6.226	2.178 ± 1.730	0.92	-	3.163 ± 3.473	2.938 ± 2.163	0.75	-
LTX A4 5S, 6R	0.000 ± 0.000	0.000 ± 0.000	-	-	0.058 ± 0.271	0.000 ± 0.000	0.52	-
LTX A4 5S, 6R, 15R	0.028 ± 0.046	0.008 ± 0.027	0.05 *	0.53	0.015 ± 0.025	0.032 ± 0.049	0.58	-
Leukotriene B4	0.027 ± 0.015	0.026 ± 0.015	0.61	-	0.026 ± 0.012	0.027 ± 0.014	0.98	-
15 HETE	0.348 ± 0.267	0.256 ± 0.162	0.08 **	0.4	0.232 ± 0.105	0.342 ± 0.240	0.33	-
12 HETE	1.733 ± 1.058	1.660 ± 1.151	0.43	-	1.871 ± 1.272	1.688 ± 0.835	0.92	-
5 HETE	0.030 ± 0.018	0.024 ± 0.008	0.40	-	0.021 ± 0.009	0.025 ± 0.009	0.14	-
5 oxo ETE	0.188 ± 0.088	0.182 ± 0.074	0.63	-	0.192 ± 0.133	0.165 ± 0.095	0.83	-
16 HETE	0.032 ± 0.118	0.000 ± 0.000	0.26	-	0.000 ± 0.000	0.000 ± 0.000		-
EPA metabolites								
18 HEPE	0.113 ± 0.031	0.105 ± 0.042	0.29	-	0.100 ± 0.029	0.123 ± 0.057	0.23	-
Resolvin E1	0.039 ± 0.027	0.088 ± 0.153	0.45	-	0.058 ± 0.094	0.048 ± 0.040	0.50	-
DHA metabolites								
Protectin D1	0.060 ± 0.091	0.036 ± 0.041	0.72	-	0.034 ± 0.043	0.056 ± 0.055	0.58	-
NPD1	0.141 ± 0.094	0.105 ± 0.059	0.31	-	**0.097 ± 0.085**	**0.140 ± 0.075**	**0.06 ****	**0.53**
Resolvin D1	0.172 ± 0.199	0.097 ± 0.057	0.55	-	0.238 ± 0.378	0.164 ± 0.243	0.53	-
Maresin 1	0.032 ± 0.019	0.028 ± 0.019	0.34	-	0.030 ± 0.009	0.036 ± 0.016	0.33	-

LTX, lipoxin; HETE, hydroxyeicosatetraenoic acid; HODE, hydroxyoctadecadienoic acid, Avg—mean, SD—standard deviation, * statistically significant, ** at the limit of significance (treated as a trend).

**Table 3 biomedicines-11-00092-t003:** Eicosanoid levels in smokers and non-smokers, including gender.

Eicosanoids (ng/mL)	Smokers		*p*-Value	Cohen’s d	Non-Smokers		*p*-Value	Cohen’s d
	Women	Men			Women	Men		
	Avg ± SD n = 16	Avg ± SD n = 11			Avg ± SD n = 24	Avg ± SD n = 22		
LA metabolites								
9 HODE	0.023 ± 0.017	0.041 ± 0.054	0.28	-	0.036 ± 0.019	0.032 ± 0.024	0.31	-
13 HODE	0.027 ± 0.02	0.041 ± 0.057	0.64	-	0.032 ± 0.020	0.033 ± 0.024	0.57	-
AA metabolites								
TXB2	0.072 ± 0.047	0.086 ± 0.045	0.34	-	0.105 ± 0.072	0.046 ± 0.023	0.01 *	1.10
Prostaglandin E2	2.178 ± 1.730	2.938 ± 2.163	0.42	-	4.101 ± 6.226	3.163 ± 3.473	0.93	-
LTX A4 5S, 6R	0.000 ± 0.000	0.000 ± 0.000	-	-	0.000 ± 0.000	0.058 ± 0.271	0.32	-
LTX A4 5S, 6R, 15R	0.008 ± 0.027	0.032 ± 0.049	0.14	-	0.028 ± 0.046	0.015 ± 0.025	0.45	-
Leukotriene B4	0.026 ± 0.015	0.027 ± 0.014	0.98	-	0.027 ± 0.015	0.026 ± 0.012	0.72	-
15 HETE	0.256 ± 0.162	0.342 ± 0.240	0.42	-	0.348 ± 0.267	0.232 ± 0.105	0.03 *	0.57
12 HETE	1.660 ± 1.151	1.688 ± 0.835	0.54	-	1.733 ± 1.058	1.871 ± 1.272	0.90	-
5 HETE	0.024 ± 0.008	0.025 ± 0.009	0.44	-	0.030 ± 0.018	0.021 ± 0.009	0.13	-
5-oxo ETE	0.182 ± 0.074	0.165 ± 0.095	0.82	-	0.188 ± 0.088	0.192 ± 0.133	0.64	-
16 HETE	0.000 ± 0.000	0.000 ± 0.000	-		0.032 ± 0.118	0.000 ± 0.000	0.18	-
EPA metabolites								
18 HEPE	0.105 ± 0.042	0.123 ± 0.057	0.36	-	0.113 ± 0.031	0.100 ± 0.029	0.29	-
Resolvin E1	0.088 ± 0.153	0.048 ± 0.040	0.98	-	0.039 ± 0.027	0.058 ± 0.094	0.90	-
DHA metabolites								
Protectin D1	0.036 ± 0.041	0.056 ± 0.055	0.53	-	0.060 ± 0.091	0.034 ± 0.043	0.94	-
NPD1	0.105 ± 0.059	0.140 ± 0.075	0.27	-	**0.141 ± 0.094**	**0.097 ± 0.085**	**0.07 ****	**0.46**
Resolvin D1	0.097 ± 0.057	0.164 ± 0.243	0.64	-	0.172 ± 0.199	0.238 ± 0.378	0.97	-
Maresin 1	0.028 ± 0.019	0.036 ± 0.016	0.12	-	0.032 ± 0.019	0.030 ± 0.009	0.80	-

LTX, lipoxin; HETE, hydroxyeicosatetraenoic acid; HODE, hydroxyoctadecadienoic acid, Avg—mean, SD—standard deviation, * statistically significant, ** at the limit of significance (treated as a trend).

## Data Availability

The data will be made available on request.

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
