# Peer review of "Smoking Affects the Post-Stroke Inflammatory Response of Lipid Mediators in a Gender-Related Manner"

_biomedicines, 2022, doi:10.3390/biomedicines11010092_

Round 1
Reviewer 1 Report
This study aimed to evaluate the post-stroke inflammatory response. The concentration of selected derivatives of arachidonic acid, linoleic acid, DHA, and EPA, which may be markers of post-stroke inflammation were detrmined. The results of the tests were compared between the groups depending on sex and smoking status.
MAJOR COMMENTS:
1. In the abstract, there are numerous errors in the symbol> specifying the value of p (instead of <)
2. In sentence "Generally, men and women, smokers showed a much smaller amount of correlations than non-smokers" According to the authors, is the amountr of correlations the most important, not their strength and statistical significance?
3. In my opinion, the abstract lacks a clear indication of the conclusions of the research.
4. Was the incidence of inflammatory diseases among the exclusion criteria? If not, whether their occurrence was compared between the analyzed groups in order to assess their influence on the obtained results.
5. In Table 1, the numbers have too many decimals. These are not important values and affect the readability of the results.
6. In Table 1, twice the numbers 0.00 are given as p values. I suggest using p <0.05 or p <0.01 etc.
7. In subsection "Isolation and determination of eicosanoids" the sentence "The following relationships were analyzed" is not clear to me.
8. In addition, despite providing references, I suggest providing basic information on the methodology for determining the concentrations of the studied mediators (calibration curve, internal standard method?).
9. For me, the description of the results of the correlation analysis is redundant. I suggest shortening it because this information is visible on the chart. It's hard to read.
10. Units are missing from the tables (except table 1). What are the
values given (mean, median)? If it is mean and standard deviation, have
parametric tests been used everywhere? in this type of data, there are
often deviations from the assumptions of parametric tests and relying on
the mean and standard deviation may lead to drawing erroneous conclusions.
For statistically significant differences, it is worth adding scatterplots
with plotted raw data.
Author Response
Dear Reviewer,
We would like to thank You for your valuable comments regarding our manuscript "Smoking affects the post-stroke inflammatory response of lipid mediators in a gender-related manner". All their comments and suggestions were taken into account, which increased the value of the articles. All changes made to the manuscript are marked in red.

Reviewer 2 Report
The idea of the manuscript is very interesting but there are some aspects that must be addressed before publication
1) English expressions and scientific vocabulary. A native speaker or a professional company must review the text. Some examples:
Line 16: "we compared..." Avoid personal verb forms
Line 32: Idem
Line 110-113: Rewrite (sentence is too long)
2) Delete Table 1. It is not relevant for the study. Or at least display a short table with the general characteristics of the sample without statistical analysis (it is not relevant)
3) The intro
- It is too long
- " < and >" symbols are wrong
4) Introduction. Add the aims at the end
5) Methods and Results
- Improve presentation and format of table 3.
- Add the analisys of the effect size (Cohen's)
6) Add limitations of the study such as confounding variables, factors that might have affected the results of the study.......
Author Response

(The authors gave the same response as above.)

Reviewer 3 Report
[Major concern]
It is hard to follow and I did not really understand the justification for repeating the study on smoking affecting the post-stroke inflammatory response.
“Li, B.; Li, D.; Liu, J.-F.; Wang, L.; Li, B.-Z.; Yan, X.-J.; Liu, W.; Wu, K.; Xiang, R.-L. “Smoking Paradox” Is Not True in Patients 511 with Ischemic Stroke: A Systematic Review and Meta-Analysis. J Neurol 2021, 268, 2042–2054, doi:10.1007/s00415-019-512 09596-3.”
“Pan, B.; Jin, X.; Jun, L.; Qiu, S.; Zheng, Q.; Pan, M. The relationship between smoking and stroke: A meta-analysis. Medicine 517 2019, 98, e14872, doi:10.1097/MD.0000000000014872.”
Compared to the well-written meta-analyses, what can be newly known?
Furthermore, the sample size was too small to conclude.
[Minor concern]
73 human subjects were included. Was this study approved by any institutional review board? Please provide its number.
The imbalance of subjects in each group could affect the findings of this study. What was the complementary method in the analysis?
Author Response

(The authors gave the same response as above.)

Round 2
Reviewer 1 Report
The authors corrected the manuscript in an appropriate, careful manner. In my opinion, it's fit for publishing after tweaking a few minor details (listed below):
Repahse:
(abstract) . This research highlights how much smoking changes inflammation in both genders with the participation of lipid mediators, making smoking cessation undeniable. - This study highlights the effect of smoking on inflammation in both genders mediated by lipid mediators, which makes smoking cessation undeniable (changing the word "changes" to a more appropriate one).
(2.2) and solvent B was 157 methanol/water/acetic acid (100/0/0.1, v/v). ./vol./vol.) - and solvent B was 157 methanol/water/acetic acid (100/0/0.1, v/v/v) -
(3.2.) Statistically significant dependencies - Statistically significant relationships
(units) ml - mL